# Strongyloidiasis in Auckland: A ten-year retrospective study of diagnosis, treatment and outcomes of a predominantly Polynesian and Fijian migrant cohort

Tim Cutfield[1]*, Soana Karuna Motuhifonua[2], Matthew Blakiston[3], Hasan Bhally[4], Eamon Duffy[2,5], Rebekah Lane[5], Erik Otte[6], Terri Swager[7], Amanda Maree Taylor[2], Veronica Playle[1,2,7]

1 Department of Infectious Diseases, Te Whatu Ora Counties Manukau, Auckland, New Zealand, 2 Faculty of Medical and Health Sciences, The University of Auckland, Park Road, Grafton, Auckland, New Zealand, 3 Department of Microbiology, Labtests Auckland, Mount Wellington, Auckland, New Zealand, 4 Department of Infectious Diseases, Te Whatu Ora Waitematā, Takapuna, Auckland, New Zealand, 5 Department of Infectious Diseases, Te Whatu Ora Te Toka Tumai Auckland, Grafton, Auckland, New Zealand, 6 Department of Microbiology, Canterbury Health Laboratories, Hagley Avenue, Christchurch Central City, Christchurch, New Zealand, 7 LabPLUS Auckland, Auckland City Hospital, Grafton, Auckland

* tim.cutfield@middlemore.co.nz

**Data Availability Statement:** Ethical approval for this study by Auckland Health Research Ethics

## Abstract

### Background

*Strongyloides stercoralis* is not endemic in Aotearoa New Zealand (AoNZ). However, approximately one third of Auckland residents are born in endemic countries. This study aimed to describe the epidemiology and management of strongyloidiasis in Auckland, with a focus on migrants from Pacific Island Countries and Territories.

### Methods

This study retrospectively reviewed clinical, laboratory and pharmacy records data for all people diagnosed with strongyloidiasis in the Auckland region between July 2012 and June 2022. People with negative *Strongyloides* serology were included to estimate seropositivity rate by country of birth.

### Findings

Over ten years, 691 people were diagnosed with strongyloidiasis. Most diagnoses were made by serology alone (622, 90%). The median age was 63 years (range 15–92), 500 (72%) were male, and the majority were born in Polynesia (350, 51%), Fiji (130, 19%) or were of Pasifika ethnicity (an additional 7%). Twelve participants (1.7%) had severe strongyloidiasis at diagnosis. The total proportion treated with ivermectin was only 70% (484/691), with no differences between immunocompromised and immunocompetent participants, nor by ethnicity. The outcome of treatment (based on a combination of serology and/or eosinophilia and/or stool microscopy) could only be determined in 50% of the treated cohort. One participant failed treatment with ivermectin, experiencing recurrent strongyloidiasis, and

Committee does not currently permit open sharing of this study's de-identified data set. The criteria for a waiver of consent to be granted, in accordance with the New Zealand National Ethics Committee Standards 2019 (points 7.46 to 7.48), include a specific requirement for data protection. Unfortunately as the ethics application for this study did not include a request for open data sharing, the authors are not currently permitted to share the de-identified data set with a data sharing repository. Study data may possibly be made available after a request addressed to the AHREC (ahrec@auckland.ac.nz).

**Funding:** This work was supported by a Summer Studentship grant, awarded to SKM by Te Whatu Ora Counties Manukau. The funder had no role in the study design, data collection, data analysis, data interpretation, writing of the report, nor decision to submit for publication.

**Competing interests:** The authors have declared that no competing interests exist.

another participant died in association with severe strongyloidiasis. The rate of 'positive' *Strongyloides* serology was highest among participants born in Samoa (48%), Fiji (39%), and Southeast Asian countries (34%).

## Interpretation

Strongyloidiasis was common and under-treated in Auckland during the study period. Clinicians should have a low threshold for considering strongyloidiasis in migrants from endemic countries, including Polynesia and Fiji.

## Author summary

*Strongyloides stercoralis* is endemic to most tropical and subtropical countries, and causes potentially life-long infection of the gastrointestinal tract. Strongyloidiasis can occasionally cause life-threatening illness, particularly in immunocompromised people, and often years after the initial infection. Very little is known about the burden of strongyloidiasis in Oceania, particularly in the Pacific Island Countries and Territories of Polynesia and Fiji. Auckland (New Zealand) is home to a large population of migrants from Polynesia and Fiji, and clinicians in Auckland frequently treat strongyloidiasis.

This study describes a cohort of 691 people diagnosed with strongyloidiasis in Auckland between 2012 and 2022. Only 70% of people in this cohort received treatment. The proportion receiving treatment did not differ significantly among people with the highest risk of severe strongyloidiasis. In addition, high rates of positive *Strongyloides* serology were found among people born in Samoa and Fiji, which were comparable to seropositivity among migrants from other 'high burden' countries (South East Asia). In addition to identifying a need to improve management of strongyloidiasis in Auckland, this study suggests the burden of strongyloidiasis in Polynesia and Fiji may be higher than previously suggested.

## Introduction

Strongyloidiasis is an infection caused by the soil-transmitted helminth *Strongyloides stercoralis*. *Strongyloides stercoralis* is endemic in tropical and subtropical regions globally, with estimates of the global burden of infection ranging from 614 million to 1.2 billion [1,2]. However, strongyloidiasis is of global importance due to its unique life cycle, whereby autoinfection of the host facilitates indefinite infection, even after migration to non-endemic settings [3]. Most people with strongyloidiasis are asymptomatic or have mild non-specific symptoms such as gastrointestinal discomfort, diarrhoea, or urticaria [4]. Chronic eosinophilia is common, but not universal (48–78%) and may be intermittent [5]. Immunocompromised people are at risk of severe strongyloidiasis, including hyperinfection syndrome. Even modestly immunocompromised states have been associated with severe strongyloidiasis, including diabetes, cirrhosis, alcohol dependence, and as little as a single dose of systemic corticosteroid [6–8]. Severe strongyloidiasis is associated with acceleration of the *S. stercoralis* life-cycle, resulting in high numbers of filariform larvae invading the gastrointestinal tract and disseminating to distant organ systems [3,7]. Even with treatment, severe strongyloidiasis is associated with a mortality of up to 63% [7]. Evidence-based guidelines recommend ivermectin treatment for all people diagnosed with strongyloidiasis [9]. Ivermectin is an inexpensive, well-tolerated and effective

(rates of cure 86–97%) agent for the treatment for chronic strongyloidiasis [9]. A single dose is non-inferior to a four-dose regimen for immunocompetent people with non-severe disease [10].

Although considered endemic for *S. stercoralis*, there are no overall estimates for the burden of strongyloidiasis in Oceania [11]. Aotearoa New Zealand (AoNZ) is not endemic for *S. stercoralis* [12]. However, in AoNZ's largest city, Auckland (population 1.6 million), approximately 32% of the population were born in endemic regions, primarily in Asia or in the Pacific Island countries and territories of Fiji and Polynesia (mostly Samoa, Tonga, the Cook Islands and Niue Island) [13]. Despite clinicians in Auckland frequently managing strongyloidiasis, published data from AoNZ to date have been limited to descriptions of infections acquired during international deployment, identified during routine refugee screening, and case reports in migrants from Polynesia or Fiji [8,14–19].

This study aimed to describe a cohort of people with strongyloidiasis and their management in Auckland, AoNZ. Additionally, this study aimed to improve understanding of the burden of strongyloidiasis in Polynesia and Fiji by describing the rate of positive *Strongyloides* serology tests by country of birth. Lastly, it was hoped that the results of this study might contribute to improving awareness and management of strongyloidiasis in the Auckland region.

## Methods

### Ethics

Ethical approval was granted for this study by the Auckland Regional Health Research Ethics Committee (AHREC; ref AH25100). Additional approvals were obtained for all participating Te Whatu Ora Health New Zealand districts: Waitematā, Te Toka Tumai Auckland, Counties Manukau and Waitaha Canterbury (Canterbury Health Laboratories). In accordance with National Ethics Advisory Committee standards 2019, a waiver of informed consent was granted as the study required only such data as was collected during routine clinical care, obtaining retrospective consent would be prohibitively impractical, the benefits of being included in the study were likely to outweigh any risks of inclusion, and password-protected data would be stored securely on the Counties-Manukau server.

### Setting and study design

This was a retrospective descriptive study of strongyloidiasis, including management, diagnosed in both community and inpatient settings in the Auckland region over a ten-year period (1st July 2012 and 30th June 2022). Data collection took place from December 2022 and May 2023, with a minimum 6-month follow-up period after diagnosis for all participants. From 2000 to 2022, three geographically distinct organisations provided healthcare in the Auckland region: Waitematā, Auckland and Counties-Manukau District Health Boards (DHBs). Following changes to AoNZ's health system in 2022, these organisations were designated as districts of Te Whatu Ora Health New Zealand.

### Participants and data collection

Participants with strongyloidiasis were identified from three data sources. Firstly, electronic records from two laboratories (Labtests and LabPLUS, respectively responsible for all Auckland regional community and in-hospital parasite identification) were searched for samples with microscopy-confirmed *S. stercoralis*. Both laboratories use microscopy with formalin ethylacetate concentration for detection of *S. stercoralis* larvae in stool.

Secondly, participants who had *Strongyloides* serology testing were identified from regional electronic health records. During the study period, *Strongyloides* serology testing was performed at Canterbury Health Laboratories (Christchurch, AoNZ) using the Bordier *Strongyloides ratti* commercial IgG ELISA assay (Bordier Affinity Products, Crissier, Switzerland). The estimated performance characteristics of this assay as reported by the manufacturer include a diagnostic sensitivity of 90% and specificity of 70–90%. Other estimates vary and are described elsewhere [20,21]. For this assay an optical density (OD) ratio of $> 1.2$ is considered 'positive', $< 0.9$ 'negative', and 0.9–1.2 'equivocal'. Seventeen participants diagnosed prior to 2017 had *Strongyloides* serology test performed by Pathology West, Westmead (New South Wales, Australia) using an in-house *Strongyloides ratti* IgG ELISA [22].

Thirdly, participants with a hospital discharge diagnosis code for strongyloidiasis (International Statistical Classification of Diseases, 10th revision (ICD- 10) codes: B78.0, B78.1, B78.7, B78.9) were identified from the Auckland regional data warehouse. These participants were included if, after review of clinical records, an alternative laboratory method of *S. stercoralis* infection was made (e.g., on histology).

Participants were excluded if: strongyloidiasis was diagnosed outside of Auckland, no electronic health records were available, they were less than 1 year old at diagnosis, there was no laboratory confirmation of *S. stercoralis* infection, or if they were diagnosed as part of routine refugee screening at the Mangere Refugee Resettlement Centre [17].

Participants with strongyloidiasis were subclassified as 'positive' (microscopy-confirmed strongyloidiasis, or 'positive' serology, or both) or 'equivocal' (equivocal serology, and no positive microscopy). We elected to include participants with equivocal serology as clinicians may interpret 'equivocal' serology as sufficient evidence for treatment in some scenarios (e.g., prior to significant immunocompromise). Participants with 'negative' serology and no microscopic identification of *S. stercoralis* were considered '*Strongyloides* negative' and were included for an analysis of rate of positive *Strongyloides* serology by region of birth.

## Procedures

Demographic data (age, sex, ethnicity, DHB of residence) were obtained from the regional data warehouse. All clinical data, as well as country of birth (if missing from the information management system), were obtained by review of electronic clinical records. Eosinophilia prior to treatment was defined as any eosinophil count above $0.5\times10^9$/L within 5 years of diagnosis. Clinical variables associated with risk of severe strongyloidiasis, including HTLV-1 infection, dispensing of systemic corticosteroid within 6 months of diagnosis, alcohol dependence, diabetes or immunocompromise, were recorded. Immunocompromise was defined as the presence of medication-associated immunocompromise (*Table A in S1 Text*), an immunocompromising condition (solid organ transplant, haematologic malignancy, solid organ malignancy with chemotherapy within 3 months of diagnosis, or autoimmune chronic inflammatory disease requiring immunosuppression) or both. Ivermectin dispensing, including date, dose and duration of prescription was obtained from both hospital and community pharmacy dispensing records. Severe strongyloidiasis was defined by either a) the presence of a hospital discharge summary coded diagnosis of severe strongyloidiasis or *Strongyloides* hyperinfection, or b) records confirming a clinical syndrome of severe strongyloidiasis (e.g., colitis, pneumonitis, typical cutaneous findings of hyperinfection, gram-negative meningitis, or gram-negative sepsis with no alternative diagnosis) in association with microscopy-confirmed *S. stercoralis* infection. The clinical records of all people who died during the study period were reviewed. Deaths were considered 'definitely' associated with strongyloidiasis if preceded by severe strongyloidiasis. Possible strongyloidiasis-associated death was defined as

death following a syndrome consistent with severe strongyloidiasis, but without confirmatory microscopy.

## Outcome measures

The primary outcome was the proportion of participants with strongyloidiasis treated with ivermectin. Given the absence of regional consensus guidance for treatment, we defined 'treatment' as evidence of at least one dose of ivermectin dispensed after a diagnosis of strongyloidiasis. Other treatments are not publicly funded in AoNZ for treatment of strongyloidiasis, so were not assessed. Secondary outcomes included: a) the proportion of important subgroups who received ivermectin: NZ Māori and Pasifika compared with other ethnicities, and immunocompromised compared with immunocompetent people; b) the proportion of participants who had successful treatment of strongyloidiasis (see definitions below); c) the proportion of participants with severe strongyloidiasis; d) the rate of positive *Strongyloides* serology by country or region of birth.

## Definitions of treatment outcome

Definitions of treatment outcome were adapted from those in the 'STRONG 1–4' trial [10]. Treatment outcome was subcategorised as *definite* success (all three of stool microscopy conversion, serologic response, and resolution of eosinophilia), *probable* success (resolution of eosinophilia *or* serologic response, with or without confirmed stool microscopy conversion) and *possible* success (stool microscopy conversion, but without serologic response or resolution of eosinophilia). Stool microscopy conversion was defined as negative stool microscopy only, from more than 1 week after completion of treatment in participants with pre-treatment stool microscopy-confirmed strongyloidiasis. Serologic response was defined as at least a 50% reduction in IgG ELISA OD value following treatment. Resolution of eosinophilia was defined if all eosinophil levels more than one month after treatment were below $0.5 \times 10^9$/L. Treatment failure was defined if *S. stercoralis* larvae were identified in a clinical sample more than 1 week after completion of treatment. All other outcomes were considered as 'treatment outcome not determined', e.g., if no post-treatment serology testing nor stool microscopy were requested, and eosinophilia persisted or relapsed.

## Statistical analysis

Analyses were performed using 'R' v4.22 (The R Foundation for Statistical Computing, https://www.r-project.org). Descriptive statistics were performed for demographic and clinical variables. The primary outcome and binary secondary outcomes were calculated as proportions of the total cohort with strongyloidiasis. Differences between the proportions treated in important sub-populations (described above) were evaluated using Fisher's exact test. We performed a post-hoc multiple logistic regression analysis to evaluate the potential association between the proportion treated with ivermectin and ethnicity, while accounting for potential confounding factors of age, sex, country of birth (AoNZ vs other country), immunocompromised state and equivocal vs positive strongyloidiasis diagnosis. A second post-hoc analysis compared demographic and clinical variables between people with 'positive' and 'equivocal' diagnoses, using the Wilcoxon rank sum test to assess the variable 'age', and Fisher's exact test for other variables. Lastly, an analysis of absolute numbers and rates of 'positive', compared with 'equivocal' and 'negative' *Strongyloides* serology results, by country and region of birth was performed. Sankey diagram was constructed using an online opensource tool available at SankeyMATIC.com.

### Role of the funding source

This work was supported by a Summer Studentship grant, awarded to SM by Te Whatu Ora Counties Manukau. Te Whatu Ora Counties Manukau had no role in the study design, data collection, data analysis, data interpretation, writing of the report, nor decision to submit for publication.

## Results

Between 1<sup>st</sup> July 2012 and 30<sup>th</sup> June 2022, 95 people with microscopy-confirmed *S. stercoralis*, 664 people with positive or equivocal *Strongyloides* serology (out of 2727 serology tests performed), and 92 people with an ICD-10 discharge coded diagnosis of strongyloidiasis were identified. After review, exclusion, and deduplication, this gave a total of 691 people with strongyloidiasis meeting the inclusion criteria, including 584 'positive' and 107 'equivocal' diagnoses (*Fig 1*). Participants with 'equivocal' diagnoses had similar rates of comorbidity to those with 'positive' diagnoses, but were less likely to be born in PICT (55% vs 72%), had a younger mean age (54 vs 64 years) and a lower proportion were male (56% vs. 75%) (*Table B in S1 Text*). The majority of participants (622, 90%) were diagnosed with strongyloidiasis on the basis of 'positive' or 'equivocal' serology alone (*Fig 2*). The Auckland region's annual requests for *Strongyloides* serology increased from 78 in 2017, to 720 in 2020. Consequently, most participants were diagnosed after 2019 (405/691; *S1 Fig*).

The pre-treatment characteristics of the 691 participants with strongyloidiasis are described in *Table 1*. The median age was 63 years (range 15–92 years) and 500 (72%) were male. Seventy percent of participants were born in Polynesia (350/691, 51%) or Fiji (130/691, 19%). An additional 51 (7%) participants not born in Pacific Island countries or territories were of Pasifika ethnicity. Overall, 70% (486/691) of participants had at least one risk factor for developing severe strongyloidiasis, with diabetes and systemic corticosteroid use the most common (*Table 1*).

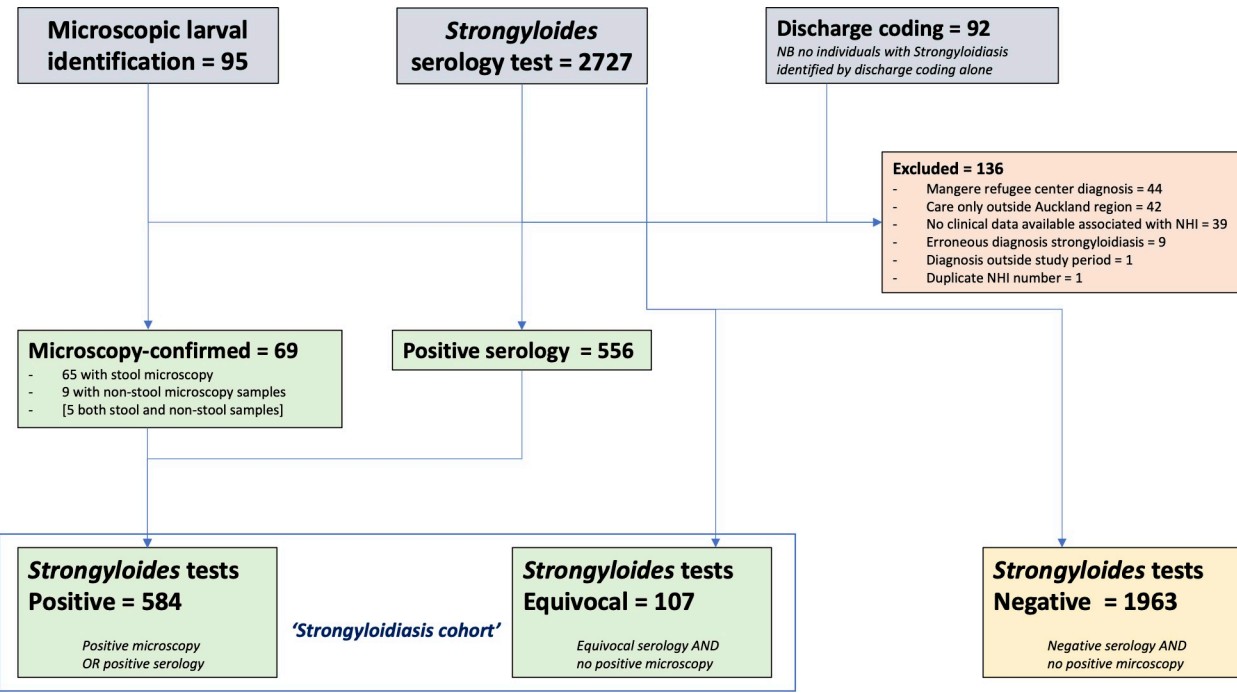

**Fig 1. Consort diagram of data sources and derivation of strongyloidiasis cohort (n = 691).**

| | Eosinophilia@ | Microscopy Positive* | Microscopy Negative | Microscopy Not performed | Serology Totals |
|---|---|---|---|---|---|
| Serology Positive | Yes | 41 | 144 | 323 | 556 |
| | No | 0 | 12 | 36 | |
| Serology Negative | Yes | 2 | 0 | 0 | 2 |
| | No | 0 | 0 | 0 | |
| Serology Equivocal | Yes | 1 | 22 | 53 | 108 |
| | No | 0 | 3 | 29 | |
| Serology Not performed | Yes | 22 | 0 | 0 | 25 |
| | No | 3 | 0 | 0 | |
| Microscopy Totals | | 69 | 181 | 441 | Total cohort = 691 |

**Fig 2. Interaction between three pre-treatment laboratory components of strongyloidiasis diagnosis: microscopic identification of larvae,** *Strongyloides* **serology result, and presence of eosinophilia (n = 691).** * Includes 60 participants with only stool positive, 5 with stool and other sample positive, and 4 with only other sample positive. @ Overall 608 / 691 (88%) of individuals diagnosed with strongyloidiasis had preceding eosinophilia.

## Pre-treatment diagnostic characteristics of strongyloidiasis

Stool microscopy was performed in 250 (36%) of 691 participants during evaluation for strongyloidiasis. *S. stercoralis* larvae were also identified in an additional 9 non-stool samples. Overall, 69 (10%) participants had microscopy-confirmed strongyloidiasis including 60 from stool samples, five from both stool and non-stool samples, and four from only non-stool samples. The nine non-stool samples were gastrointestinal tract tissue samples obtained by endoscopic biopsy (7 samples), bronchoalveolar lavage (1 sample), and urine specimen (1 sample) (*Fig 2*).

*Strongyloides* serology was performed in 666 (96%) of 691 participants with strongyloidiasis. Serology results were 'positive' in 556 (83%) and 'equivocal' in 108 (16%). Serology was falsely negative in two profoundly immunocompromised participants (0.3%) with microscopy-confirmed severe strongyloidiasis: one solid organ transplant recipient, and one with rheumatoid arthritis treated with rituximab and leflunomide. One immunocompetent participant with 'equivocal' serology had positive stool microscopy. The majority of participants (622, 90%) were diagnosed with strongyloidiasis on the basis of serology alone, including 441 (64%) without requesting stool microscopy (*Fig 2*).

Eosinophilia was present in 608 (88%) participants prior to diagnosis.

## Severe strongyloidiasis and deaths

Twelve participants (1.7%) had severe strongyloidiasis at diagnosis. Eleven of these had risk factors for severe strongyloidiasis, including six with recent systemic corticosteroid use and

**Table 1. Pre-treatment characteristics of strongyloidiasis cohort (n = 691).**

| | |
|---|---|
| **Age (years),** median (range) | 63 (15, 92) |
| **Male** n (%) | 500 (72%) |
| **Region of birth** n (%) | |
| Pacific Island Countries and Territories (PICT) | 481 (70%) |
| Polynesia | 350 (51%) |
| Samoa | 246 (36%) |
| Tonga | 55 (8%) |
| Cook Islands | 38 (5%) |
| Other Polynesia[1] | 11 (2%) |
| Fiji | 130 (19%) |
| Other PICT[2] | 1 (0.1%) |
| Aotearoa New Zealand | 88 (13%) |
| Asia | 77 (11%) |
| Southeast Asia | 33 (5%) |
| Indian Subcontinent | 24 (4%) |
| Other Asian Country | 20 (3%) |
| Africa | 10 (1%) |
| Other | 13 (2%) |
| Not available | 22 (3%) |
| **Ethnicity** n (%) | |
| NZ Māori | 29 (4%) |
| Other | 203 (29%) |
| Pasifika | 459 (66%) |
| **District Health Board** n (%) | |
| CMDHB | 347 (50%) |
| ADHB | 205 (30%) |
| WDHB | 139 (20%) |
| **Immunocompromised** | 68 (10%) |
| **Diabetes** | 381 (55%) |
| **Alcohol dependence** | 25 (4%) |
| **Corticosteroid use within 6 months** | 188 (27%) |
| **HTLV-1** | |
| Negative | 4 (1%) |
| Not tested | 687 (99%) |

[1] Other Polynesia: Niue Island (4), Tuvalu (4), American Samoa (2), Tahiti (1)

[2] Other Pacific Island Countries and Territories (PICT): Kiribati (1)

three that were immunocompromised. One participant without pre-determined risk factors for severe strongyloidiasis, but was malnourished, had *Strongyloides*-associated colitis and gram-negative sepsis. Severe strongyloidiasis directly contributed to the death of one participant, and a second died following an illness compatible with severe strongyloidiasis, but without confirmatory microscopy testing. Another participant without severe strongyloidiasis died of surgical complications after excision of a periampullary lesion that was postoperatively confirmed as *Strongyloides*-associated on histology. An additional 113 (16%) participants died from causes apparently unrelated to strongyloidiasis during the follow up period.

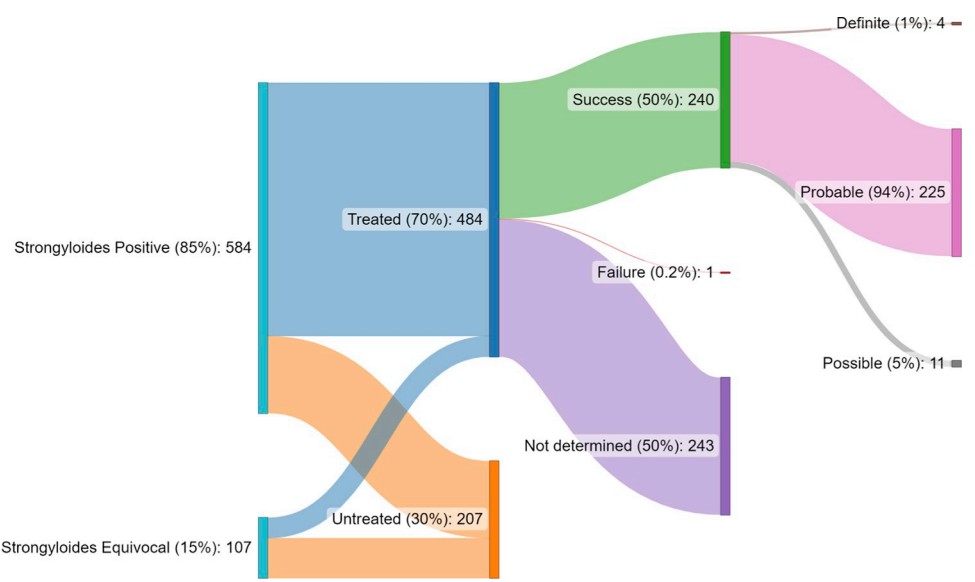

**Fig 3. Sequential proportions of strongyloidiasis cohort by 'positive' and 'equivocal' diagnosis, ivermectin treatment received, and treatment outcomes, including sub-categorisation of 'treatment success' (n = 691).**

## Treatment of strongyloidiasis and outcome

Evidence of ivermectin dispensing was available for 484 (70%) of 691 participants with strongyloidiasis (*Fig 3*). The proportion treated with ivermectin was higher for participants with 'positive' diagnoses (447/584, 77%) than 'equivocal' diagnoses (37/107, 35%; p<0.01). The median dose of ivermectin prescribed was 18mg (range 3mg to 33mg), given for a median of 2 doses (IQR 1–2, range 1–20).

Post-treatment ('test of cure') *Strongyloides* serology was performed in 20% (98/484) of treated participants. Serology was performed at least 6 months after treatment in 63 (64%) of 98 participants. Post-treatment stool microscopy was performed in 63 (13%) of 484 participants. Post-treatment eosinophilia assessment was available for 94% (457/484) participants.

Treatment success was confirmed for 240 (50%) participants, 225 (94%) of whom were sub-classified as 'probable' (resolution of eosinophilia *or* serologic response, with or without confirmed stool microscopy conversion) treatment success (*Fig 3*). The outcome of treatment was not determined in 243 participants (50%). There was one (0.2%) confirmed treatment failure in an immunocompromised participant who was receiving high-dose prednisone and cyclophosphamide for an ANCA-associated vasculitis. This participant experienced two symptomatic, stool microscopy-confirmed relapses that required treatment over the subsequent six months. The two relapses occurred after two-dose ivermectin treatment (days 1 and 14), with probable success following four-dose treatment (days 1,2, 14 and 15).

## Treatment of Māori, Pasifika and immunocompromised participants

There was no significant difference in the proportion treated with ivermectin between immunocompromised (75%) and immunocompetent participants (70%) (p = 0.4). On univariate analysis a significantly lower rate of treatment was observed for NZ Māori (48%) compared with Pasifika (71%) and other ethnicities (70%) (p = 0.036). However, there was no association between ethnicity and the likelihood of receiving treatment in a post-hoc multiple logistic regression model (*Table 2*). In this model, receipt of ivermectin was significantly associated

**Table 2. Multiple logistic regression model of likelihood of ivermectin prescription by ethnicity, adjusted for potential confounding factors (n = 691).**

|  | OR | 95% CI | p-value |
|---|---|---|---|
| **Ethnicity** |  |  |  |
| Other ethnicity (ref) | *ref* | - | - |
| NZ Māori | 0.78 | 0.29 – 2.11 | 0.63 |
| Pasifika | 0.87 | 0.58 – 1.29 | 0.50 |
| **Age** | 1.00 | 0.99 – 1.01 | 0.87 |
| **Country of birth** |  |  |  |
| Aotearoa New Zealand | *ref* | - | - |
| Other country | 2.04 | 1.11 – 3.71 | 0.02 |
| Not available | 1.64 | 0.56 – 5.34 | 0.4 |
| **Sex** |  |  |  |
| Female | *ref* | - | - |
| Male | 1.03 | 0.69 – 1.52 | 0.88 |
| **Immunocompromised** | 1.38 | 0.76 – 2.62 | 0.30 |
| **Strongyloidiasis diagnosis** |  |  |  |
| Equivocal | *ref* | - | - |
| Positive | 6.10 | 3.85 – 9.80 | <0.001 |

with being born outside of AoNZ (OR 2.04, 95% CI 1.11–3.71, p = 0.02) and having a 'positive' rather than 'equivocal' diagnosis of strongyloidiasis (OR 6.10, 95% CI 3.85–9.80, p <0.001).

## Rate of positive *Strongyloides* serology by country of birth

Country or region of birth was available for 97% (669/691) of the strongyloidiasis cohort and 96% (1889/1963) of the '*Strongyloides* negative' cohort. Twenty-five (4%) of 691 people in the strongyloidiasis cohort did not have *Strongyloides* serology performed and were excluded from this analysis. Rates of 'positive', 'equivocal' and 'negative' strongyloidiasis serology by country or region of birth are displayed in *Table 3*. The highest rates of 'positive' strongyloidiasis serology were in participants born in Samoa (48%), Fiji (39%) and Southeast Asia (34%). The rate of 'equivocal' strongyloidiasis serology was similar between different regions of birth (3–9%). Participants born in AoNZ had a low rate of 'positive' serology (9%).

**Table 3. Proportion of positive, equivocal and negative *S. stercoralis* serology among participants with pre-treatment serology testing, by region and country of birth (n = 2629).**

|  | *S. stercoralis* serology result | | |
|---|---|---|---|
|  | Positive (n = 559) | Equivocal (n = 107) | Negative (n = 1,963) |
| **Region of birth** |  |  |  |
| Aotearoa New Zealand | 65 (9%) | 22 (3%) | 681 (89%) |
| Pacific Island Countries and Territories (PICT) |  |  |  |
| Samoa | 219 (48%) | 25 (6%) | 211 (46%) |
| Fiji | 108 (39%) | 10 (4%) | 157 (57%) |
| Tonga | 42 (20%) | 11 (5%) | 159 (75%) |
| Cook Islands | 27 (26%) | 9 (9%) | 70 (66%) |
| Other PICT* | 8 (11%) | 4 (6%) | 61 (84%) |
| Asia |  |  |  |
| Southeast Asia | 26 (34%) | 3 (4%) | 47 (62%) |

*(Continued)*

**Table 3.** (Continued)

| | S. stercoralis serology result | | |
| --- | --- | --- | --- |
| | Positive (n = 559) | Equivocal (n = 107) | Negative (n = 1,963) |
| Indian Subcontinent | 17 (10%) | 7 (4%) | 147 (86%) |
| Other Asian Country[&] | 13 (8%) | 7 (4%) | 150 (88%) |
| Africa[#] | 4 (5%) | 4 (5%) | 66 (89%) |
| Other countries[@] | 9 (6%) | 4 (3%) | 140 (92%) |
| Not available | 21 (22%) | 1 (1%) | 74 (77%) |

*Other Pacific Island Countries and territories (PICT): American Samoa (7), Kiribati (6), Niue (35), Papua New Guinea (1), Tahiti (4), Tokelau (2), Tuvalu (18)

[&]Other Asian Country: China (People's Republic of) (60), Hong Kong (11), Indonesia (5), Japan (4), South Korea (Republic of Korea) (9), Malaysia (14), Philippines (57), Singapore (5), Taiwan (5)

[#] African countries: Algeria (1), Burundi (2), Democratic Republic of Congo (1), Egypt (1), Eritrea (1), Ethiopia (14), Ghana (1), Kenya (2), Namibia (1), Nigeria (2), Somalia (10), South Africa (25), Sudan (1), Tanzania (1), Zambia (2), Zimbabwe (9)

[@]Other countries: Afghanistan (16), Australia (13), Azerbaijan (1), Brazil (1), Canada (2), Chile (5), Colombia (2), Croatia (2), Denmark (1), England (35), France (1), Germany (2), Great Britain (13), Greece (1), Guyana (1), Iran (5), Iraq (11), Ireland (1), Israel (1), Italy (1), Kuwait (2), Latvia (1), Macedonia (1), Mexico (4), Netherlands (5), Portugal (1), Russia (3), Saudi Arabia (1), Scotland (5), Slovakia (1), Spain (2), Switzerland (1), Syria (5), USA (2), Uruguay (1), Yugoslavia (2)

## Discussion

This study describes the demographics, evaluation, treatment and follow-up of people diagnosed with strongyloidiasis in Auckland over a ten-year period. Thirty percent of participants had no evidence of treatment with ivermectin. Concerningly, the low treatment proportion also extended to immunocompromised participants; 25% of whom were not treated. Although treatment rates in similar contemporary studies have ranged from 71%-97%, international guidance strongly recommends treatment of all people with strongyloidiasis [5,9,23,24]. As such, interventions to improve the management of strongyloidiasis in Auckland are needed.

The low proportion of treated participants reported in this study might have been contributed to by under-ascertainment of ivermectin dispensing. However, based on observations made during data collection, the following contributions are hypothesised to be more plausible explanations for undertreatment of this cohort: a) abnormal serology results not being electronically communicated from laboratories to the responsible clinician, b) abnormal results not actioned by the requesting team, c) deferral of responsibility to arrange treatment to another clinician (e.g., primary care provider), and d) misplaced confidence in the negative predictive value of stool microscopy. These hypotheses might suggest that improved clinician education and system-level interventions should be the focus of initial regional interventions to improve the management of strongyloidiasis.

Only 50% of treated participants had sufficient post-treatment follow-up to allow assessment of their treatment outcome. This was largely due to low rates of post-treatment *Strongyloides* serology testing (20%). The importance of routine post-treatment 'test of cure' might be debated for immunocompetent asymptomatic people with low-burden chronic strongyloidiasis, in whom the likelihood of cure with ivermectin is approximately 90% [9]. However, there was a high prevalence of risk factors for severe strongyloidiasis (70%) and low rate of stool microscopy at diagnosis (36%) in this cohort. As such, clinicians should be careful not to discount the importance of post-treatment 'test of cure' in the absence of data to support such an approach. Furthermore, we identified episodes of both clinically significant treatment failure and of false-negative serology in this cohort, which support the recommendation that immunocompromised people undergo multi-modal evaluation for strongyloidiasis diagnosis (e.g., serology and stool microscopy) as well as post-treatment 'test of cure' [5,9].

This study described rates of positive *Strongyloides* serology by country of birth in a clinical cohort, which may not be representative of rates of strongyloidiasis in the wider population. Nevertheless, this cohort provides new, indirect insight into the burden of strongyloidiasis in Polynesia and Fiji. The majority of people in this study were born in either Polynesia (350/691, 51%) or Fiji (130/691, 19%). Rates of 'positive' *Strongyloides* serology were highest among people born in Samoa (48%) and Fiji (39%). Previous estimations of the population burden of strongyloidiasis in Fiji and Polynesia are limited.[3,22,25] Stool surveys in Fiji have mostly reported low rates of *S. stercoralis* carriage, ranging from 0.1% to 5% [25–27]. Similarly, a 1955 survey in a Samoan village identified *Strongyloides* larvae in 6 (2.9%) of 210 stool samples [28]. However, similar to this current study, a recent report from Hawaii described a *Strongyloides* seropositivity rate of 26% (14/53) among 'Polynesian' people ('Samoan or Tongan peoples') [24]. In further support of a high burden of strongyloidiasis in Samoa and Fiji was a prevalence of 50% in a 1968 stool survey of a Fijian village, and case reports of *Strongyloides* hyperinfection in migrants to AoNZ [8,18,19,26]. Evidence of *Strongyloides* endemicity in Polynesia aside from Samoa is very limited, but includes post-deployment seroconversion in a AoNZ Police worker posted to Pitcairn [15]. Via country of birth data, this study also suggests *S. stercoralis* endemicity in the Polynesian countries/territories of American Samoa, Niue, Tuvalu and Tahiti.

There are some caveats to the rate of strongyloidiasis by country of birth reported by this study. Firstly, it is possible that some positive *Strongyloides* serology results among migrants from Polynesia and Fiji are due to cross-reactivity with other endemic nematode infections [21]. Secondly, as the indication for *S. stercoralis* testing was not collected, the seropositivity rates represent overall practice in Auckland during the study period, and could be expected to vary between clinical scenarios with different pre-test probabilities. Thirdly, the investigators had insufficient resources to collect clinical data from the 1963 people with 'negative' *Strongyloides* serology. As such, it is possible that important clinical differences might exist between the 'positive' and 'negative' *Strongyloides* serology groups. Fourthly, acquisition of *S. stercoralis* infection from a region distinct from country of birth cannot be excluded, although is thought unlikely to explain the high rates across a large cohort. Conversely, it is likely that the AoNZ-born participants diagnosed with strongyloidiasis in this cohort were infected overseas, but lack of available travel history precludes confirmation of this hypothesis. Lastly, as the participants in this study had a median age of 63 years, the high rate of strongyloidiasis described in this study may reflect an historic, rather than contemporary burden of strongyloidiasis in Polynesia and Fiji, which may have reduced in recent decades following mass drug administration programmes.

The authors are aware of one case of fatal severe strongyloidiasis diagnosed post-mortem that was not identified by this study's methodology. As such, the severe cases described in this study should be considered a minimum estimate of incidence. It is also likely that the characteristics of this cohort were influenced by clinicians' bias towards considering and testing for strongyloidiasis in restricted clinical scenarios. Such bias is suggested by the high prevalence of eosinophilia in this cohort (88%) compared with other studies (48–78%) and that treatment was significantly associated with being born outside AoNZ (OR 2.04, 95% 1.11–3.71, p = 0.02) [5]. The impact of such diagnostic bias could plausibly lead to misdiagnosis of severe strongyloidiasis in patients who do not fit the 'typical' diagnostic heuristic described in this study (e.g., older Polynesian migrants with eosinophilia). To reduce bias, future guidance should emphasise the limitations of available diagnostic modalities, the importance of obtaining a detailed travel history, and the non-specific clinical features of severe strongyloidiasis.

The findings of this study are anticipated to stimulate collaborative development and dissemination of regional guidance for strongyloidiasis. Such guidance should recommend that

people born in Polynesia and Fiji should be considered at high risk of strongyloidiasis. People with strongyloidiasis who were born in AoNZ should be the focus of future study to exclude autochthonous infection by careful consideration of previous international travel and multi-modal diagnostic evaluation to reduce the likelihood of false-positive results. Finally, contemporary assessment of the seroprevalence of strongyloidiasis in Polynesia and Fiji are warranted, which should be combined with assessment of other important endemic nematode infections.

## Conclusion

This cohort study describes the characteristics and management of strongyloidiasis in Auckland, AoNZ over a ten-year period. The total proportion of participants who received ivermectin treatment was only 70%, which falls significantly short of the standards proposed by international guidance. This study also found a high rate of serologically diagnosed strongyloidiasis in migrants from Polynesia and Fiji. As nearly 32% of people in Auckland were born in a country endemic for *S. stercoralis*, it is important that a wide range of practitioners in AoNZ are aware of how to assess for and manage strongyloidiasis. It is anticipated that development and dissemination of a collaborative regional guideline, informed by this study, will improve outcomes for people with strongyloidiasis in Auckland.

## Supporting information

**S1 Fig. Strongyloidiasis diagnoses per year of study period (July 2012 –June 2022, n = 69)***.
* *Strongyloides* serology testing became available in Aotearoa New Zealand in 2017 at Canterbury Health Laboratories (CHL). The total numbers of *Strongyloides* serology tests for the Auckland region at CHL were: 78 (2017), 331 (2018), 445 (2019), 515 (2020), 720 (2021), 524 (2022, to June).
(TIF)

**S1 Text.** Table A. Definitions of medication-associated immunosuppression. ***Table B. Comparison of pre-treatment characteristics of strongyloidiasis cohort by 'positive' or 'equivocal' diagnosis (n = 691).*** [1] Other Polynesia: Niue Island (4), Tuvalu (4), American Samoa (2), Tahiti (1). [2] Other Pacific Island Countries and Territories (PICT): Kiribati (1). [3] Wilcoxon rank sum test (age); Fisher's Exact test (other variables).
(DOCX)

## Acknowledgments

We are extremely grateful to the Information analysts, Clinical Pharmacists and Laboratory Scientists for their assistance with extracting data from electronic records: Albert Chin, Dinesh Gunaratne, Nicola Arroll, Monique Green, Natasha Pool, Nicola Davies and Rodger Linton.

## Author Contributions

**Conceptualization:** Tim Cutfield, Matthew Blakiston, Hasan Bhally, Eamon Duffy, Rebekah Lane, Erik Otte, Veronica Playle.

**Data curation:** Tim Cutfield, Soana Karuna Motuhifonua, Eamon Duffy, Terri Swager.

**Formal analysis:** Tim Cutfield, Amanda Maree Taylor.

**Funding acquisition:** Tim Cutfield.

**Investigation:** Tim Cutfield, Soana Karuna Motuhifonua, Matthew Blakiston, Hasan Bhally, Erik Otte, Terri Swager, Veronica Playle.

**Methodology:** Tim Cutfield, Soana Karuna Motuhifonua, Matthew Blakiston, Eamon Duffy, Rebekah Lane, Erik Otte, Terri Swager, Veronica Playle.

**Project administration:** Tim Cutfield.

**Supervision:** Tim Cutfield, Veronica Playle.

**Validation:** Tim Cutfield, Soana Karuna Motuhifonua.

**Writing – original draft:** Tim Cutfield.

**Writing – review & editing:** Tim Cutfield, Soana Karuna Motuhifonua, Matthew Blakiston, Hasan Bhally, Eamon Duffy, Rebekah Lane, Erik Otte, Terri Swager, Amanda Maree Taylor, Veronica Playle.

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
