## [Decision Letter · Decision Letter 0]

2 Jan 2024

Dear Dr Cutfield,

Thank you very much for submitting your manuscript "Strongyloidiasis in Auckland: a ten-year retrospective study of diagnosis, treatment and outcomes of a predominantly Polynesian and Fijian migrant cohort" for consideration at PLOS Neglected Tropical Diseases. As with all papers reviewed by the journal, your manuscript was reviewed by members of the editorial board and by several independent reviewers. In light of the reviews (below this email), we would like to invite the resubmission of a significantly-revised version that takes into account the reviewers' comments. 

The authors are encouraged to revise the paper, with particular attention paid to the comments made by Reviewer 1, then resubmit for further evaluation.

We cannot make any decision about publication until we have seen the revised manuscript and your response to the reviewers' comments. Your revised manuscript is also likely to be sent to reviewers for further evaluation.

Sincerely,

Richard Stewart Bradbury, PhD

Academic Editor

Francesca Tamarozzi

Section Editor

The authors are encouraged to revise the paper, with particular attention paid to the comments made by Reviewer 1, then resubmit for further evaluation.

Reviewer's Responses to Questions

**Key Review Criteria Required for Acceptance?**

**Methods**

-Are the objectives of the study clearly articulated with a clear testable hypothesis stated?

-Is the study design appropriate to address the stated objectives?

-Is the population clearly described and appropriate for the hypothesis being tested?

-Is the sample size sufficient to ensure adequate power to address the hypothesis being tested?

-Were correct statistical analysis used to support conclusions?

-Are there concerns about ethical or regulatory requirements being met?

Reviewer #1: 1. Line 157-162: readers would be interested in knowing how representative the study population is; authors should report the coverage of the population by the datasets (do they capture all persons seen in any clinic/hospital in the Aukland region, or just part of them?). 

2. Line 164-171: I wonder if authors can provide information on the indication of serological testing. Pretest probability would be different between patients who were tested for clinical concerns, compared with those tested for screening purposes without clinical suspicion. 

3. Line 186-188: I can understand the authors’ rationale here, but I am not certain if “equivocal” cases should be treated in the same way as others, as inclusion of these patients can skew some of analyses. Perhaps clearer distinction should be made between positive and equivocal cases, although authors try to do so to some extent in the manuscript. For example, it is probably not always wrong to not treat cases with equivocal results, whereas no treatment for positive cases is probably not good. 

4. Line 193: define “DHB”.

5. Line 202-204: how well does this approach capture ivermectin dispensing? Is there a possibility that availability of ivermectin could affect treatment, and providers could prescribe albendazole instead, even though it is less efficacious?

Reviewer #2: The aims of the study are clearly articulated, including the focus on migrants from Pacific Islands and territories.The study design is well described, logical, and appropriate to meet the aim. The population has been clearly described with the information available to the researchers, noting the appropriate exclusion criteria. Three district health services broadened the geographic areas of study in the Auckland region, and the retrospective data was collected for a 10 year period. Ethics approval is documented.

Reviewer #3: The methods are described in adequate detail (see comments regarding the serological assays)

Study design, study population and sample sizeis appropriate for a retrospective descriptive study 

Statistical analysis is appropriate

No ethical concerns

**Results**

-Does the analysis presented match the analysis plan?

-Are the results clearly and completely presented?

-Are the figures (Tables, Images) of sufficient quality for clarity?

Reviewer #1: 1. Line 268: is there anything special about before and after 2019? This is not mentioned in Discussion. Is it because of higher prevalence of strongyloidiasis after 2017-2019, more immigrants in this time period, overall increased testing, or more testing is populations with higher prevalence? Readers would at least be interested in how many serological testing was done in each year. 

2. Line 329-337: how many of these patients were equivocal? As above in Methods #3, the populations with equivocal diagnosis and positive diagnosis are likely quite different, which is supported by the Line 335-337. These populations should likely be analyzed separately. 

3. Line 344-346, and Discussion 384-385: note that this study is different from universal screening. Thus, the “positive” rate is different from seroprevalence. The significance of this data depends on pretest probability (ie as in Methods #2 above, whether there was clinical suspicion before testing). 

4. Line 357-365: this is technically a result, and should not appear for the first time in Discussion. It is currently rather anecdotal. Is there anyway authors could quantify this, or provide description in a way that is scientifically sound? 

5. Line 367-378: authors endorse test of cure strongly. It is employed in many studies to facilitate outcome assessment, but in clinical practice, as authors state, it is controversial. It is unclear how problematic this result is, and it is also unclear if this is important enough to be included in Abstract. Authors should consider evaluating clinical failure (though authors provide some anecdotes).

Reviewer #2: The novel results are clearly and completely presented. The data collection over 10 year period identified 691 persons meeting the inclusion criteria. The analysis followed the logical analysis plan as described in methods. The inclusion of clinical complications of serious strongyloidiasis was appropriate and provided further evidence of the importance of disseminating the results of this study. 

The Tables are clear - with one minor recommendation for Table 3.

Reviewer #3: The results, figures and tables are clearly presented (see comments below)

**Conclusions**

-Are the conclusions supported by the data presented?

-Are the limitations of analysis clearly described?

-Do the authors discuss how these data can be helpful to advance our understanding of the topic under study?

-Is public health relevance addressed?

Reviewer #1: To support the conclusion, I think equivocal vs positive cases should be separated. Limitations – see above in Methods and Results.

Reviewer #2: The conclusions are supported by the data presented. The limitations are clearly described and addressed questions that i had when reading the results section. The authors have provided the data that highlights the need to raise awareness for clinicians, and develop 'collaborative regional guideline' with policy makers. The epidemiological data for Pacific Islands has important public health relevance.

Reviewer #3: The conclusions are supported by the data; limitations of the study and significance of their findings are discussed adequately.

**Editorial and Data Presentation Modifications?**

Reviewer #1: Consider making Introduction more concise by providing information that is necessary to set the stage for the study only.

Reviewer #2: Minor modification: please check table 3 last column on negative tests. total is 1,963, but adds up to 1,965. Other countries shows 152 total in description, with 155 total in row in table. (The PICT, Other Asian and African countries match the table).

Reviewer #3: (No Response)

**Summary and General Comments**

Reviewer #1: This is a manuscript describing a retrospective study on strongyloidiasis in New Zealand. Authors report that strongyloidiasis was common and tended to be undertreated. Strongyloidiasis in general is understudies, and more data are needed from different parts of the world. See comments in other sections.

Reviewer #2: This impressive manuscript is an important novel contribution to epidemiological data on strongyloidiasis in Oceania. Publication of the epidemiological data will also have significant implications for including strongyloidiasis in public health strategies in Polynesia and Fiji, and beyond. This manuscript is exceptional in clarity with a logical flow of writing, application of methods and data collection and analysis. 

It has been a pleasure to review this comprehensive, concise manuscript that addresses strongyloidiasis, and i hope it will reach a wide audience. The authors anticipated action for a collaborative regional guideline is to be applauded.

Reviewer #3: Summary

In this 10-year retrospective cohort study of strongyloidiasis in Aotearoa/New Zealand the authors evaluated the diagnosis, treatment and outcomes of strongyloidiasis in a group of predominantly Polynesian and Fijian migrants residing in Auckland. The study cohort has inherent demographic biases since participant selection is non-random, compromising the generalisability of the conclusions– for example, it is unclear if the sources uncovered every Strongyloides-infected individual, and we are not provided comparative overall immigration demographic data to determine if immigrant populations from Pacific nations to AoNZ are similar in composition to the study cohort. The salient findings in this study were the high rate of eosinophilia in the Strongyloides-infected subgroup, which probably reflects selection bias by clinicians (acknowledged by the authors) and that only 70% of those that fulfilled the criteria for strongyloidiasis received ivermectin therapy. Significant shortcomings of this study include the lack of follow up data to confirm response to therapy (only 50% had available post-treatment data), lack of information on co-infecting parasites (that could cause false positivity due to cross-reactions in the ELISA) and absence of information on re-infections and from the uninfected subgroup. 

Nevetheless, the findings reported here provide evidence for a higher-than-expected prevalence of strongyloidiasis in a poorly studied region and call for the development of guidelines for detection and management of this neglected infection in the region. 

Specific comments and suggestions

Line 146 – Were there any differences in the quality of data (and conclusions drawn) from the three distinct healthcare providers mentioned? Was there any evidence of bias between them?

Line 167-9 – Since serological testing was the sole diagnostic methodology ins uhc a lare proportion of the cohort (64%) it would be useful to provide the accepted sensitivity, specificity and accuracy of the assays at the selected cutoff OD (in addition to reference to the previous publications

Line 234 – A 50% reduction in OD in the ELISA test was defined as a serological response. Why was this level of changed selected? The decrease is OD is notoriously variable, unlike a change in titres of antibodies, is variable and, for example, in the in-house assay used by WestPath, defined as positive for OD>0.25 (https://doi.org/10.1371/journal.pntd.00056070). 

Line 311 – What doses of ivermectin were used in the cases of severe strongyloidiasis?

Line 314 – Post-treatment “test of cure” serology was performed in only 20% of cases. How was the outcome of treatment determined in the remaining cases? 

Line 474 – In Figure 2, the row totals for serology positive and negative subgroups should be added to the Eosinophilia column

PLOS authors have the option to publish the peer review history of their article (what does this mean?). If published, this will include your full peer review and any attached files.

Reviewer #1: No

Reviewer #2: Yes: Wendy Ann Page

Reviewer #3: No
---

## [Editor Report · Decision Letter 1]

5 Mar 2024

Dear Dr Cutfield,

We are pleased to inform you that your manuscript 'Strongyloidiasis in Auckland: a ten-year retrospective study of diagnosis, treatment and outcomes of a predominantly Polynesian and Fijian migrant cohort' has been provisionally accepted for publication in PLOS Neglected Tropical Diseases.

Best regards,

Richard Stewart Bradbury, PhD

Section Editor

Francesca Tamarozzi

Section Editor

---

## [Editor Report · Acceptance letter]

25 Mar 2024

Dear Dr Cutfield,

We are delighted to inform you that your manuscript, "Strongyloidiasis in Auckland: a ten-year retrospective study of diagnosis, treatment and outcomes of a predominantly Polynesian and Fijian migrant cohort," has been formally accepted for publication in PLOS Neglected Tropical Diseases.

Best regards,

Shaden Kamhawi

co-Editor-in-Chief

Paul Brindley

co-Editor-in-Chief
